# Implications of VOC Oxidation in Atmospheric Chemistry: Development of a Comprehensive AI Model for Predicting Reaction Rate Constants

Xin Zhang<sup>1,2#</sup>, Jiaqi Luo<sup>1,2#</sup>, Wenxiao Pan<sup>1,2</sup>, Qiao Xue<sup>1,2</sup>, Xian Liu<sup>1,2\*</sup>, Jianjie Fu<sup>1,2,3\*</sup>, Aiqian Zhang<sup>1,2,3</sup>, Guibin Jiang<sup>1,2,3</sup>

<sup>1</sup>State Key Laboratory of Environmental Chemistry and Ecotoxicology, Research Center for Eco-Environmental Sciences, Chinese Academy of Sciences, Beijing 100085, P. R. China

<sup>2</sup> College of Resources and Environment, University of Chinese Academy of Sciences, Beijing 100190, P. R. China

<sup>3</sup> School of Environment, Hangzhou Institute for Advanced Study, University of Chinese Academy of Sciences, Hangzhou 310012, P. R. China

#These authors contributed equally to this work and should be considered co-first authors.

Correspondence to: Xian Liu (xianliu@rcees.ac.cn); Jianjie Fu (jifu@rcees.ac.cn)

Abstract. Volatile Organic Compounds (VOCs) significantly influence global atmospheric chemistry through oxidative reactions with oxidants. These reactions produce key precursors to the formation of atmospheric fine particulate matter (PM<sub>2.5</sub>) and ozone (O<sub>3</sub>), which in turn play a crucial role in regulating O<sub>3</sub> pollution and reducing PM<sub>2.5</sub> concentrations. With the increasing diversity of VOCs, the need for advanced modeling techniques to accurately estimate the atmospheric oxidation reaction rate constants ( $k_i$ , where  $i \in \{OH, Cl, NO_3, or O_3\}$ ) has become more urgent. Here we introduce Vreact, a Siamese message passing neural networks (MPNN) architecture that jointly models VOC–oxidant reactivity. The model simultaneously predicts  $log_{10}k_i$  values and achieves a mean squared error (MSE) of 0.299 and a coefficient of determination (R<sup>2</sup>) of 0.941 on the internal test set. This framework overcomes the single-oxidant constraint of traditional models, enabling unified and scalable prediction of VOC oxidation kinetics across multiple oxidants. An interactive web tool (http://vreact.envwind.site:8001) is provided to facilitate non-expert access to reactivity screening. Vreact offers valuable insights into the formation and evolution of atmospheric pollutants, and serves as a critical resource for developing effective control and emission strategies, ultimately supporting global efforts to mitigate air pollution and improve public health.

## 25 1 Introduction

The rapid advancement in data-driven methodologies has revolutionized various fields, such as protein structure prediction (Abramson et al., 2024), molecular generation (Zhang et al., 2023), organic reaction prediction (Burés and Larrosa, 2023), and bioinformatics (Theodoris et al., 2023). Environmental challenges, particularly those associated with atmospheric chemistry and climate change (Chen et al., 2024; Kubečka et al., 2023; Qiu et al., 2023; Zhao et al., 2025), have also benefited from these innovations. As pollutants evolve under both anthropogenic and natural influences, the understanding of their chemical and physical properties has become increasingly vital for addressing global air quality and climate issues. Volatile Organic

Compounds (VOCs) are organic chemicals that readily vaporize at ambient temperature, contributing significantly to the complexity of atmospheric processes. Sources of VOCs are both natural and anthropogenic, with human activities such as industrial production, petrochemical processing, and vehicle exhaust contributing to the emission of a variety of VOCs. Additionally, biosphere sources, such as plants and forests, release compounds like isoprene and monoterpenes, which further complicate atmospheric VOC dynamics (Qin et al., 2021; Sindelarova et al., 2014). These highly reactive VOCs drive critical atmospheric reactions, such as the formation of ozone and secondary organic aerosols (SOA), and significantly contribute to environmental pollution. For instance, VOCs interact with nitrogen oxides (NO<sub>x</sub>) and radicals to form tropospheric O<sub>3</sub> and SOA (Finlayson-Pitts and Pitts, 1997; Hallquist et al., 2009; Han et al., 2018; Zhang et al., 2020; Ziemann and Atkinson, 2012). The role of VOCs in the formation of secondary pollutants such as PM<sub>2.5</sub> (Huang et al., 2014; Zhao et al., 2015) and O<sub>3</sub> is a growing concern due to the adverse impacts on human health (Kamarrudin et al., 2013), including respiratory diseases, cardiovascular conditions, and overall mortality. The dynamic interactions between VOCs and atmospheric oxidants determine the persistence and transformation of these pollutants, which in turn influence their contribution to global haze, photochemical smog, and acid deposition.

VOCs undergo degradation and removal from the troposphere through diverse mechanisms driven by atmospheric oxidants. During the daytime, OH radicals serve as the primary oxidants, facilitating rapid VOC oxidation. At night, however, the concentration of OH decreases sharply due to the lack of photochemical reactions, shifting the dominant oxidation pathways to NO<sub>3</sub> radicals and O<sub>3</sub>. The reaction rates of VOCs with OH are approximately 30 times faster than those with NO<sub>3</sub> radicals, significantly influencing the spatial and temporal variation of the atmosphere's self-cleaning capacity and the formation of organic aerosols (Palmer et al., 2022; Zha et al., 2023). For example, regions with high isoprene concentrations often reflect differences in its reaction products and rates with OH and NO<sub>x</sub> rather than solely high emissions (Wells et al., 2020). Additionally, the structural diversity of VOCs determines their reaction mechanisms, influencing reaction rates. Highly reactive compounds such as alkenes, multi-substituted aromatics, and phenols exhibit higher reaction rates, whereas alkanes, alkyl nitrates, and ketones demonstrate relatively low reactivity (Ziemann and Atkinson, 2012). These variations underscore the significance of atmospheric oxidation reaction rates as key indicators of the persistence of organic pollutants in the atmosphere. Accurate assessment of these rates is essential for understanding the fate of VOCs, elucidating SOA formation processes, and addressing global challenges related to PM<sub>2.5</sub> and ozone development.

Given their importance, accurately predicting the atmospheric oxidation rates of VOCs is critical for understanding their persistence, transformation, and contribution to secondary pollutant formation. Traditionally, such predictions have relied on experimental kinetic modeling methods and computational methods (e.g., quantum-chemistry (QC) and quantitative structure-activity relationship (QSAR) approaches) (Basant and Gupta, 2018; Liu et al., 2021). Experimental methods involve tracking reactant and product concentrations using techniques like chemical ionization mass spectrometry (CIMS), followed by kinetic fitting to determine Arrhenius parameters (Logan, 1982; Wells et al., 1996). However, these methods are time-consuming and cover only a narrow subset of atmospheric VOCs. QC approaches use density-functional theory calculations such as transition-state theory (TST) or variational TST to obtain temperature-dependent rate constants (Canneaux et al., 2014; Liu et al., 2021;

65

Meana-Pañeda et al., 2024). While QC methods offer detailed mechanistic insight, their computational cost scales steeply with molecular size and conformational complexity, limiting routine application to large numbers of VOCs, However, traditional computational methods have shortcomings such as high computational complexity and low efficiency. As a more scalable alternative, QSAR models leverage molecular descriptors and statistical learning, and it has become one of the important methods for evaluating reaction rate constants. Previous examples include AOPWIN™ module integrated in US EPI Suite™ software, which applies Partial Least Squares (PLS) regression to 109 gas-phase reactions with hydroxyl radicals (Atkinson, 1986, 1987; Kwok and Atkinson, 1995), and later expansions using a broader dataset (Öberg, 2005). Some models have also incorporated machine learning algorithms such as multiple linear regression (MLR) (Liu et al., 2020, 2022) for predicting reactions with NO<sub>3</sub> and OH and artificial neural networks for predicting reactions with O<sub>3</sub> (Fatemi, 2006). Despite their utility, these models generally rely on predefined descriptors and are typically limited to reactions with a single type of oxidant, which constrains the scalability of the model. Recent advances in deep learning (DL), particularly graph neural networks (GNN), have improved molecular representation by learning features directly from molecular graphs. This enables more flexible and accurate prediction of chemical properties without requiring predefined descriptors. GNNs have been successfully applied in atmospheric chemistry and other fields tasks, such as in predicting vapor pressures with GC2NN (Krüger et al., 2025) and modeling reaction rate constants involving with OH using GAT-GIN hybrid architectures (Huang et al., 2024). However, like traditional models, these GNN-based frameworks have been developed for single-molecule systems and thus fall short in capturing the complexity of multi-molecule reactions in real environments. In contrast, the atmosphere involves competing and sequential reactions between VOCs and multiple oxidants—OH, NO<sub>x</sub>, Cl, and O<sub>3</sub>—depending on time of day, region, and chemical conditions. This multiplicity underscores the urgent need for models that can simultaneously learn and predict VOC reactivity across multiple oxidants. To meet this need, message passing neural networks (MPNN) offer a powerful framework (Gilmer et al., 2017). MPNNs propagate information across molecular graphs, capturing both atomic-level features and topological context. Extensions of MPNN, such as the communicative GraphRXN (Li et al., 2023) and directed MPNN Chemprop (Heid et al., 2024), have shown promise in learning reactivity across multiple reactants. Compared with the simple concatenation using molecular fingerprints/descriptors, they all use MPNN to deeply extract task-relevant representations of chemical reactions, provide abundant chemical information for subsequent reaction modeling, and achieve good prediction results. Yet, their application has largely focused on synthesis or materials chemistry, not atmospheric oxidation reaction. This study addresses this gap by proposing Vreact, a novel Siamese MPNN architecture capable of jointly modeling reactions between VOCs and four major atmospheric oxidants. Unlike previous models that treat each oxidant independently, Vreact processes VOC-oxidant pairs in a unified framework, it learns representations from the molecular graphs of VOCs and oxidants through the MPNN, and encodes their interactions via feature aggregation. This design enables the model to accept arbitrary VOC-oxidant combinations and simultaneously predict reaction rate constants  $k_i$  (where  $i \in \{OH, Cl, NO_3, or O_3\}$ ). The dual-input design of Vreact enhances scalability and generalization across multiple oxidants. Ablation experiments show that Vreact significantly outperforms a structurally simpler single-input MPNN trained under identical conditions. The interaction module within Vreact provides atomic-level attention maps that offer mechanistic insights into VOC-oxidant


reactivity patterns, improving interpretability. Applying Vreact to 447 atmospheric VOCs not included in the training data revealed a wide distribution of oxidation reactivities and confirmed that alkenes and aromatics exhibit higher reactivity, acting as key precursors for ozone and SOA formation.

# 2 Methods and Data





## 2.1 Collection and Preprocessing of Reaction Rate Constant Dataset

The VOCs reaction rate constant dataset compiled by McGillen et al. is utilized in the study, which includes gas-phase reaction rate constants of natural atmospheric VOCs, halocarbons, and their degradation products with OH, Cl, NO<sub>3</sub> radicals, and O<sub>3</sub>, within a temperature range of 250-370K (McGillen et al., 2020). Under thermodynamic standard conditions at 298K, a total of 2802 gas-phase reaction rate constant data points were obtained, encompassing 1586 VOCs and 4 oxidants. This dataset includes  $k_i$  values for 1363 VOCs with OH, 735 VOCs with Cl, 393 VOCs with NO<sub>3</sub> radicals, and 311 VOCs with O<sub>3</sub>. Due to the wide range of reaction rate constants  $k_i$  in the dataset  $(1.460 \times 10^{-21} \sim 7.550 \times 10^{-10} \text{cm}^3/(\text{molecule·s})$ , S.D.=±1.040×10<sup>-10</sup>), the data were log-transformed to  $\log_{10}k_i$  to reduce skewness and mitigate the influence of outliers on the model. To ensure a balanced distribution of each type of oxidant in the training, validation, and internal test sets, the dataset was divided using stratified random sampling into training, validation and internal test sets in an 8:1:1 ratio (Table S1). Combinations of the same VOC with different oxidants may appear across the training, validation, and internal test sets.

## 2.2 Construction and Training of the Vreact Model

All VOCs and oxidant molecules were converted into graphs G(V, E) (Text S1). The generated molecular graph G includes ten types of atomic information for each non-hydrogen atom, such as element type, chirality, and atomic hybridization type, as well as four types of bond information, including bond type and conjugation (Table S2). A Siamese MPNN architecture-Vreact, was designed to simultaneously accept input features of VOCs and oxidant molecules (Fig. 1). The model takes the SMILES of VOCs and oxidants as input and primarily includes a VOC molecular graph representation layer and a MPNN layer, an oxidant molecular graph representation layer and MPNN layer, an interaction layer, and a prediction layer. The molecular graph G(V, E) encoding layers of VOCs and oxidants containing node feature matrix X and edge feature matrix Y, which learn molecular properties through the MPNN layer (Gilmer et al., 2017). The MPNN forward propagation process consists of two phases: Message Passing Phase and Readout Phase and generates molecular feature tensors A for VOCs and B for oxidants. Subsequently, the interaction layer transforms the molecular features A of VOCs and B of oxidants into tensors A1 and B1 of the same shape and concatenates them into tensor E2. Reaction rate constants are determined not only by the molecular structure of the reactants but also by the interactions between the reactants. The interaction feature tensor E1 is dot-multiplied with E2 to obtain the oxidant-affected VOC feature tensor E3, similarly, it is dot-multiplied with E4 to obtain the VOC-affected oxidant feature tensor E5. These operations embed the learned interaction features into the molecular structure features, providing a more comprehensive representation of the chemical reaction mechanisms between the two reactants. The

prediction phase is composed of a pooling layer and three fully connected layers. The pooling layer uses the Set2Set method to achieve global average pooling, and the fully connected layers map the input features to the final predicted values ( $\log_{10}k_i$ ). More details can be found in Text S2.



During model training, Adaptive Moment Estimation (Adam) (Kingma and Ba, 2017) was employed to address the fixed learning rates issue in traditional gradient descent methods. Adam adaptively adjusts the learning rate of each parameter using first-order moment estimates (mean of the gradients) and second-order moment estimates (exponentially moving average of the uncentered variance of the gradients), aiding in rapid model convergence. Bayesian optimization was utilized for hyperparameter tuning, which included the initial learning rate of the optimizer (*Ir*), batch size, L2 regularization parameter (*weight decay*), dropout rate (*p*), and MPNN time steps (*T*) (Text S3). During hyperparameter optimization, the hyperparameter combination that minimizes the Mean Squared Error (MSE) of the validation set was selected as the optimal hyperparameter combination, and the best model was saved (Table S3). The predictive performance of the model was assessed using MSE, Root Mean Square Error (RMSE), Mean Absolute Error (MAE), and coefficient of determination (R<sup>2</sup>) (Text S4). For more information on the model implementation, please refer to Text S5.

Figure 1. Schematic of the Vreact Architecture. SMILES of VOCs and oxidants are converted into molecular graphs, where nodes represent atoms and edges represent bonds. Atomic and bond features form matrices X and Y. Using a Siamese MPNN architecture, the Vreact model processes these features through separate MPNN layers for VOCs and oxidants. The final prediction layer outputs  $log_{10}k_i$ , incorporating both molecular and interaction features.

# 150 2.3 Clustering Analysis




Morgan fingerprints (radius 2, 1024 bits, generated using RDKit) was used as the molecular embeddings before clustering and visualization. To investigate VOC structural diversity and reactivity trends, two methods were applied: the Self-Organizing Map (SOM) (Kohonen, 2006) and the Uniform Manifold Approximation and Projection (UMAP). The SOM algorithm clustered VOCs into 100 structural groups (10×10 grid), using a sigma of 0.3 and learning rate of 0.5. The UMAP algorithm projected the high-dimensional fingerprint space into 2D for visualization, with the number of neighbors set to 50, minimum distance to 0.6, and metric as correlation.

## 3 Results and Discussion

## 3.1 Analysis of VOC and Oxidant Reaction Data Distribution and Characteristics

The categories and distribution characteristics of VOC and oxidant reaction data are first explored in the study, which includes log<sub>10</sub>k<sub>i</sub> data for 1586 VOCs with OH, Cl, NO<sub>3</sub>, and O<sub>3</sub> (Fig. 2A). The dataset contains the most data for OH, accounting for 48.64% of the total, as OH plays a crucial role in the atmosphere, rapidly reacting with organic pollutants and dominating their removal process. The remaining data points are for Cl (26.23%), NO<sub>3</sub> (14.03%), and O<sub>3</sub> (11.1%) in descending order of data quantity. O<sub>3</sub> is primarily produced through photochemical reactions involving NO<sub>x</sub> and VOCs, while NO<sub>3</sub>, as the principal nighttime atmospheric oxidant, significantly contributes to the oxidation and removal of trace gases. The dataset encompasses VOCs with diverse chemical structures, including 22 molecular motifs such as double bonds, esters, benzene rings, and halogen atoms (F, S, Cl, Br, and I) (Fig. 2B). This extensive chemical structure space facilitates the model's ability to learn more structural features and enhances its generalization capability.

Moreover, although there is some overlap in the reactions of the four oxidants with VOCs, each oxidant also has specific VOC reactions (Fig. 2C). There are 747 VOCs with  $k_i$  data for only one oxidant and 839 VOCs with  $k_i$  data for multiple oxidants, of which 81 VOCs have data for all four oxidants. For example, isoprene can react with OH, NO<sub>3</sub>, and Cl through hydrogen abstraction reactions, and undergo addition reactions with O<sub>3</sub> *via* its unsaturated double bonds. Furthermore, the four oxidants exhibit different  $\log_{10}k_i$  value distribution with VOCs due to differences in chemical structures and reactivity (Fig. 2D). OH, due to its high oxidation potential, usually reacts quickly with VOCs *via* hydrogen abstraction, with  $\log_{10}k_i$  concentrated in the range of -14.000 to -10.000. In contrast, O<sub>3</sub> typically undergoes slower addition reactions with unsaturated bonds in reactants (Ziemann and Atkinson, 2012), with  $\log_{10}k_i$  ranging from -20.836 to -13.721. NO<sub>3</sub> can participate in both hydrogen abstraction and addition reactions, resulting in a wider range of  $\log_{10}k_i$  values. The diverse reaction rates of these oxidants maintain the composition and oxidative state of aerosols in the atmosphere, but the uneven distribution of their values makes predicting  $k_i$  more challenging. Even for the same oxidant, VOCs with different structures exhibit varied reaction rates in gas-phase oxidation reactions. For example, NO<sub>3</sub> reacts very slowly with aromatic rings, with a  $k_i$  value of 3.900×10<sup>-16</sup> cm<sup>3</sup>/(molecule·s)

for xylene. In contrast, NO<sub>3</sub> can rapidly abstract hydrogen from hydroxyl groups, with a  $k_i$  value of up to  $1.72\times10^{-10}$  cm<sup>3</sup>/(molecule·s) for 3-methylcatechol.

Figure 2. Visualization of VOCs Dataset. (A) Proportion of the four types of oxidants. (B) Number of VOCs containing each molecular motif. MultFct: multifunctional; AroRings: aromatic rings; NaRings: non-aromatic rings; Tbonds: triple bonds; CumDBs: cumulated double bonds; ConjDBs: conjugated double bonds; SepDBs: separated double bonds. (C) Number of VOCs that can undergo oxidation reactions with the four oxidants. (D) Distribution of  $\log_{10}k_i$  values for the four oxidants. (E) Heatmap of reaction rate constants based on VOCs clustering, where each grid represents a cluster of structurally similar VOCs. The color gradient indicates the  $\log_{10}k_i$  values, with red indicating higher  $\log_{10}k_i$  values (faster reaction rates), blue indicating lower  $\log_{10}k_i$  values (slower reaction rates), and white indicating the absence of  $\log_{10}k_i$  data for that cluster. The cluster containing butyl acrylate are enclosed within the black box.

Furthermore, the same VOCs show different reaction rates with different oxidants. The SOM algorithm was used to explore the relationship between VOC structural variation and  $\log_{10}k_i$ . Each grid in Fig. 2E represents a VOC cluster, and the color gradient indicates reactivity (average  $\log_{10}k_i$  values) for the corresponding oxidants. By comparing  $\log_{10}k_i$  values across clusters, oxidant-specific reactivity patterns can be assessed. For example, butyl acrylate (CAS RN.141-32-2) reacts slowly with NO<sub>3</sub> radicals and O<sub>3</sub>, mainly due to the unsaturated addition reactions through the carbon-carbon double bond, where the ester group in the molecular structure produces an electron-withdrawing effect, reducing the electron density in the  $\pi$  bond and thus lowering the reaction rate (Gai et al., 2009; Wang et al., 2010). In contrast, it reacts faster with OH and Cl through hydrogen abstraction rather than addition (Le Calvé et al., 1997; Ohta, 1984; Wang et al., 2018). This demonstrates that the dataset, which includes various oxidants and VOCs, exhibits diverse  $\log_{10}k_i$  values. The overall  $\log_{10}k_i$  values differ significantly

between different oxidants. This diverse dataset enables the model to learn the reaction information between VOCs and different oxidants, thereby improving model performance and prediction accuracy.

# 3.2 Performance Evaluation of Vreact Model





The Siamese MPNN architecture of the Vreact captures both molecular features of VOCs and oxidants as well as their interaction dynamics simultaneously. During hyperparameter optimization, the set of hyperparameters that minimized MSE on the validation set was selected. After training for 46 epochs (Fig. S1), Vreact achieved robust predictive performance on the validation set, with  $R^2$  of 0.961, MSE of 0.194 and MAE of 0.314 for  $\log_{10}k_i$  (Fig. 3A). On the internal test set, the model achieved  $R^2$  of 0.941, MSE of 0.299 and MAE of 0.322 for  $\log_{10}k_i$  (Fig. 3A), indicating robust predictive capability and excellent generalization ability for unseen VOC-oxidant combinations. The small MAE difference between the validation set and internal test sets, despite a larger difference in MSE, indicates that MSE is more sensitive to outliers or large errors, while MAE directly reflects the average absolute prediction error. Although the  $R^2$  on the internal test set is slightly lower than on the validation set, this minor discrepancy does not affect the model's robust predictive ability. The result on the internal test set is available in Table S4.

To explore the predictive performance of the Vreact model for different types of oxidants, we evaluated the prediction performance for OH, Cl, O<sub>3</sub>, and NO<sub>3</sub> separately. The regression fit of predicted  $\log_{10}k_i$  values versus experimental values for the four oxidants (Fig. 3B) shows that O<sub>3</sub> and NO<sub>3</sub> have higher dispersion compared to OH and Cl. The R<sup>2</sup> values for the reactions of the four oxidants, in descending order, are OH > Cl > NO<sub>3</sub> > O<sub>3</sub>, with OH and Cl having R<sup>2</sup> values of 0.929 and 0.913, respectively. The prediction performance for NO<sub>3</sub> radicals and O<sub>3</sub> is comparatively lower, with R<sup>2</sup> values below 0.800. The OH dataset is the most abundant and balanced, while data amount of O<sub>3</sub> and NO<sub>3</sub> was relatively small, and the model can't fully capture the reaction features, leading to prediction bias. In addition, the  $\log_{10}k_i$  values for NO<sub>3</sub> are highly dispersed, also reducing the prediction performance. Additionally, the order of the size of R<sup>2</sup> is consistent with the order of the data volume of the four oxidant datasets. This indicates that the amount of data is also an important factor affecting the prediction performance of reaction rate constants, and that more available data help the model to fully capture reaction features.

Figure 3. Evaluation and comparison of the predictive performance of the Vreact model. (A) MSE, MAE, R<sup>2</sup> of Vreact (trained on the McGillen et al. dataset) on the validation set, internal test set, and external post-2020 test set. (B) R<sup>2</sup> values for log<sub>10</sub>k<sub>i</sub> predictions of four oxidants' reactions in the internal test set. (C) Distribution of AE between predicted and experimental log<sub>10</sub>k<sub>i</sub> values for the four oxidants in the internal test set. (D) R<sup>2</sup> of the Vreact and Vreact-Ablation on the OH, Cl, NO<sub>3</sub>, O<sub>3</sub>, and combined test sets. (E) R<sup>2</sup> comparison among previously published single-oxidant models, the original Vreact (evaluated on cleaned literature test sets), and Retrained Vreact (trained and tested using the same original splits as the literature) highlighting adaptability. (F-H) The chemical spatial distribution of VOCs in the OH, O<sub>3</sub>, and NO<sub>3</sub> datasets used in this study and prior literature.

The Absolute Error (AE) between the predicted and experimental  $log_{10}k_i$  values for the four types of oxidants are presented in Fig. 3C. The median AE for OH is 0.149, while O<sub>3</sub> and NO<sub>3</sub> exhibit median AEs of 0.301 and 0.287, respectively, which are

slightly higher than that of OH. Overall, 84% of the AE values for O<sub>3</sub> and NO<sub>3</sub> are within 1. As depicted in the Fig. 3C, individual outliers in AE contribute to the increased RMSE and MAE for O<sub>3</sub> and NO<sub>3</sub>, and the consequent decrease in R<sup>2</sup>. For example, the AE for the reaction of NO<sub>3</sub> with azulene ( $C_{10}H_8$ ) is 4.653. Azulene, an aromatic hydrocarbon composed of a seven-membered ring fused to a five-membered ring, is an isomer of naphthalene ( $C_{10}H_8$ ). NO<sub>3</sub>, as electrophilic reagents, tend to attack regions with higher electron density. Compared to naphthalene, the electron density distribution of azulene is uneven, with certain regions having high electron density that may facilitate effective interactions with NO<sub>3</sub>. Additionally, the structure of azulene may reduce steric hindrance, allowing NO<sub>3</sub> radicals easier access to reaction sites (Atkinson et al., 1992), resulting in a higher reaction rate constant and increasing the model's prediction difficulty. Similarly, the predicted  $log_{10}k_i$  value for the reaction of NO<sub>3</sub> with diiodomethane ( $CH_2I_2$ ) is significantly lower than the true value (AE=2.763). This discrepancy may be attributed to the limited representation of iodine-containing VOCs in the dataset, with only iodomethane ( $CH_3I$ ) and iodoethane ( $C_2H_3I$ ) having  $k_i$  values in the training and validation sets. This limited data prevents the model from fully learning the reaction characteristics of iodine-containing compounds, resulting in a larger prediction error for diiodomethane with NO<sub>3</sub> radicals.

## 245 3.3 Model Ablation Study






To evaluate the contribution of the Siamese neural network architecture in Vreact, we performed an ablation study. In the ablation model (Vreact-Ablation), the oxidant input and interaction module were removed, leaving only the VOC input. Both Vreact and Vreact-Ablation were trained, validated, and tested on the OH, Cl, NO<sub>3</sub>, O<sub>3</sub>, and combined datasets. All experimental settings were kept consistent, including data sources (McGillen et al., 2020), hyperparameters and evaluation metrics. As shown in Fig. 3D, Vreact consistently outperformed Vreact-Ablation across all four oxidants, with R<sup>2</sup> improvements of 0.049 (OH), 0.113 (Cl), 0.184 (NO<sub>3</sub>), and 0.021 (O<sub>3</sub>). When evaluated on the combined dataset, Vreact-Ablation achieved an R<sup>2</sup> of only 0.035, indicating that it fails to generalize across multiple oxidants. Additionally, both models showed comparable runtime per iteration. These results demonstrate that, under the same training conditions, the Siamese MPNN architecture significantly enhances predictive performance and generalization. By explicitly modeling VOC-oxidant interactions, the architecture enables the network to capture shared patterns across reaction types, thereby improving its practical applicability in multi-reactivity prediction.

#### 3.4 Comparation with Single-Oxidant Prediction Models

Most existing machine learning models for predicting VOC reaction rate constants are tailored for individual oxidants, limiting their applicability to complex atmospheric systems involving multiple oxidants. In contrast, the Siamese MPNN architecture of the Vreact enables simultaneous learning of molecular features and interaction patterns across different VOC—oxidant pairs within a unified framework. To benchmark Vreact against previously published single-oxidant QSAR/ML models, we selected three top-performing models developed under 298K conditions: Liu et al. (2020) for OH (training/test = 144/36), Xu et al. (2013) for O<sub>3</sub> (60/35), and Liu et al. (2022) for NO<sub>3</sub> radicals (151/38). Prior to evaluation, UMAP was applied to reduce the dimensionality of the Morgan molecular fingerprints to visualize the chemical space of both the comparison literature datasets

and the Vreact training set (Fig. S2). The observed structural overlap confirms that Vreact's dataset spans a broad and diverse chemical space. Given that our study used different data than those reported in the literature, we employed two strategies for comparison. First, the pre-trained Vreact model (trained on the McGillen dataset) was directly applied to the literature test sets to evaluate extrapolation performance. To ensure a fair comparison, overlapping data points between the literature test sets and the McGillen training set were removed (2 of 38 for NO<sub>3</sub>, 13 of 35 for O<sub>3</sub>, and 6 of 36 for OH). Second, Vreact was retrained on each literature dataset using their original train/test splits (Retrained Vreact), allowing a direct comparison with published models on original literature test sets.

As shown in Fig. 3E, both the original Vreact model and its retrained version consistently outperformed the single-oxidant models from Liu et al. (2022) and Xu et al. (2013) on the OH and O<sub>3</sub> literature test sets, achieving higher R<sup>2</sup> values and demonstrating superior regression fits between predicted and experimental values. These results highlight the capability of the Vreact architecture—whether trained on a broad multi-oxidant dataset or finetuned on smaller single-oxidant datasets—to effectively learn structural features of VOCs and oxidants and capture complex molecular interactions through its Siamese MPNN framework. Notably, Vreact shows opposite performance trends for OH and O<sub>3</sub> between the internal and literature test set. To understand this, UMAP was applied to project compounds from the training, internal, and literature test sets into a shared chemical space. As shown in Fig. 3F, the internal OH test set overlaps well with the training data, leading to consistently strong performance. In contrast, the literature OH set is sparse and scattered near the dataset boundaries. Despite this, Vreact still achieves a high R<sup>2</sup>, demonstrating good generalization. For O<sub>3</sub> (Fig. 3G), the internal test set lies farther from the dense training distribution, contributing to lower R<sup>2</sup>. Meanwhile, the literature O<sub>3</sub> set is better aligned with the training data, resulting in higher prediction accuracy. For NO<sub>3</sub> (Fig. 3H), both internal and literature sets show similar distributions, and the model achieves comparable R<sup>2</sup> values (~0.815). Although Vreact underperforms slightly compared to the original single-oxidant model, retraining on the literature data improves performance. This suggests that multi-oxidant training may introduce some noise but does not significantly compromise prediction accuracy.

## 3.5 Mechanism Insights Through Interaction Analysis






The interaction layer of the Vreact model can elucidate the atomic interaction mechanisms between VOCs and oxidants. The interaction matrix, sized  $n_1 \times n_2$ , where  $n_1$  represents the number of non-hydrogen atoms in the VOC molecule and  $n_2$  represents the number of non-hydrogen atoms in the oxidant molecule. Mapping these interaction coefficients onto the molecular structure highlights key atoms that determine the reaction rate.

To exemplify this mechanism, we analysed specific cases. 2-methyl-4-penten-2-ol is an unsaturated oxygenated volatile organic compound (OVOC) that constitutes a significant proportion of the atmospheric VOCs, primarily sourced from industrial solvents used in ink and jet ink manufacturing (Li et al., 2021). As shown in Fig. 4A, the interaction coefficient for the distal unsaturated carbon atoms is the highest during the reaction with O<sub>3</sub>, indicating these are likely the reaction sites for O<sub>3</sub> attack. It is inferred that O<sub>3</sub> adds to the unsaturated carbon-carbon double bond through an addition reaction, forming primary ozonides (POZs). These POZs are unstable intermediates that rapidly cleave to produce carbonyl compounds and

carbon-based radicals, which further rearrange to form secondary ozonides (SOZs). The SOZs and their reaction products are precursors of SOA. Another example is  $\gamma$ -caprolactone (GCL), a five-membered ring ester used in perfumes, which rapidly reacts and degrades with OH upon entering the atmosphere. Interaction weight analysis shows that the carbon atom linked to the ethyl group contributes most to GCL's oxidative degradation by OH (Fig. 4B), suggesting that OH initially attacks this carbon atom, abstracting a H atom to form a carbon radical. Previous studies indicate that the reactivity of carbons adjacent to the oxygen atom in lactones is particularly significant in reactions with OH, especially when alkyl substituents are attached to this carbon, which enhances its reactivity (Barnes et al., 2014).





Figure 4. Visualization of atomic weights in VOC molecules. (A) Reaction process of 2-methyl-4-penten-2-ol with O<sub>3</sub>. (B) Reaction process of  $\gamma$ -caprolactone with OH. The darker the highlighted color of the atom, the stronger its interaction in the gas-phase oxidation reaction.

## 3.6 Evaluating Extrapolation Ability and Prioritizing VOCs for Environmental Impact

To further validate the extrapolation capability and generalization performance of the Vreact model, developed using a dataset compiled up to the year 2020 (Baptista et al., 2021; Joudan et al., 2022; Li et al., 2021), additional  $k_i$  data from experimentally measured VOCs and oxidants published after 2020 were collected as an external test set (post-2020 test set) (Table 1). The prediction results showed that the AE between the experimental  $\log_{10}k_i$  and the predicted values was within 1, with the reaction rate constant prediction for γ-heptalactone and OH exhibiting the smallest prediction error. The AE for γ-heptalactone with OH was only 0.005, and the overall MAE was 0.240, with an MSE of 0.112 and an  $R^2$  of 0.98 (Fig. 3A shown in red). The results indicate that the Vreact can accurately predict the atmospheric oxidation reaction rate constants of unknown VOCs,

demonstrating its potential application in addressing complex atmospheric chemistry issues involving the interactions between VOCs and oxidants.





Table 1. The prediction results on the post-2020 test set.

| VOC name                   | Chemical structure                                     | Oxidant        | Experimental log <sub>10</sub> k <sub>i</sub> | Predicted log <sub>10</sub> k <sub>i</sub> | AE    | Ref.                    |
|----------------------------|--------------------------------------------------------|----------------|-----------------------------------------------|--------------------------------------------|-------|-------------------------|
| 2-methyl-4-<br>penten-2-ol | HO                                                     | O <sub>3</sub> | -17.370                                       | -16.712                                    | 0.658 | (Li et al.,<br>2021)    |
| γ-caprolactone             |                                                        | ОН             | -11.194                                       | -11.209                                    | 0.015 | (Baptista et al., 2021) |
|                            |                                                        | C1             | -9.886                                        | -10.149                                    | 0.263 | (Baptista et al., 2021) |
| γ-heptalactone             |                                                        | ОН             | -11.056                                       | -11.051                                    | 0.005 | (Baptista et al., 2021) |
|                            |                                                        | C1             | -9.770                                        | -9.943                                     | 0.173 | (Baptista et al., 2021) |
| FESOH                      | F <sub>3</sub> C F F F F F F F F F F F F F F F F F F F | ОН             | -11.377                                       | -11.876                                    | 0.499 | (Joudan et al., 2022)   |
|                            |                                                        | Cl             | -10.824                                       | -10.759                                    | 0.065 | (Joudan et al., 2022)   |

FESOH: 2- (1,1,2-trifluoro-2-heptafluoropropyloxy-ethylsulfanyl)-ethanol; AE: absolute error

Despite the identification of hundreds of VOC species, the environmental behavior of most VOCs in the atmosphere and their potential contributions to particulate matter formation and ozone increase remain largely unclear. To address this gap, we employed the Vreact model to evaluate the atmospheric oxidation reaction rate constants of a broad spectrum of VOCs. Molecular structures for 447 VOCs with unknown atmospheric oxidation  $k_i$  values were collected from previous research, which evaluated more than 500 Chinese domestic source profiles, including literature and field measurements (Sha et al., 2021) (Table S5). After excluding VOCs already included in the Vreact dataset, 296, 339, 416, and 369 data points for OH, Cl, O<sub>3</sub>, and NO<sub>3</sub> were retained, respectively. The prediction results indicated that, although the oxidation reaction rates of VOCs in the atmosphere vary (Fig. 5A), the differences in  $\log_{10}k_i$  values are primarily influenced by the type of oxidant, with smaller variations in  $\log_{10}k_i$  values observed for different VOCs reacting with the same oxidant. Among these, reactions with OH and Cl were the fastest, consistent with the results from the McGillen dataset analysis used in the modeling (Fig. 2D). Additionally, the changes in the proportion of VOC types within different reaction rate intervals (Fig. 5B) demonstrated that the composition of VOC types varied with reaction rates. Halocarbons exhibited relatively slower reaction rates, while alkenes and aromatics reacted relatively quickly, and oxygenated compounds showed a more uniform rate distribution. Consequently, areas with high

emissions of alkenes and aromatics will produce more reaction products per unit time, providing precursors for O<sub>3</sub> and SOA formation (Gao et al., 2021).

The top five VOCs with the fastest reaction rates with OH, Cl, O<sub>3</sub>, and NO<sub>3</sub> were further examined in the study (Fig. 5C). Among these, 2,6-Dimethyl-2,6-cyclooctadiene (CAS RN: 3760-14-3) is a volatile compound with an irritating odor, exhibiting the fastest reaction rates with OH, Cl, and O<sub>3</sub>. Additionally, 1,3-cyclopentadiene (CAS RN: 542-92-7) and 1,4-Dimethylcyclohexene (CAS RN: 70688-47-0) also showed high reaction rates with O<sub>3</sub>, Cl, and OH, likely due to the presence of double bonds and cyclic structures in these molecules. The carbon atoms in the double bonds and those connected to methyl groups generally have high reactivity. Therefore, it could be inferred that these VOCs, or VOCs with similar structures, may significantly contribute to the formation of fine particulate matter and the increase in ozone in the atmosphere.



Figure 5. Predicted reaction rate constants for VOCs atmospheric oxidation reactions. (A) Predicted mean  $\log_{10}k_i$  values for different types of VOCs. (B) Distribution of VOC types ranked by predicted reaction rates, divided into quartiles: the fastest 25% (Q1), 25%-

50% (Q2), 50%-75% (Q3), and the slowest 25% (Q4). (C) Molecular structures of VOCs with the fastest reaction rates with the four oxidants.

## 350 4 Concluding


In response to growing concerns about atmospheric pollution and its impact on human health and climate, this study introduces Vreact, a deep learning model designed to predict oxidation rate constants for VOCs with multiple oxidants (OH, Cl, O<sub>3</sub>, and NO<sub>3</sub>). Vreact demonstrates strong overall performance (MSE=0.299, R<sup>2</sup>=0.941 on internal test data) and provides mechanistic insights by capturing atomic-level interaction patterns through a Siamese MPNN framework. Its predictive accuracy varies by oxidant, reflecting the availability and diversity of training data. The model achieves high accuracy for OH (R<sup>2</sup>=0.929, n=1363) and Cl (R<sup>2</sup>=0.913, n=735), supporting robust application in daytime oxidation modeling. In contrast, lower performance is observed for NO<sub>3</sub> (R<sup>2</sup>=0.721, n=393) and O<sub>3</sub> (R<sup>2</sup>=0.584, n=311), pointing to challenges in modeling oxidants with fewer data and more complex mechanisms. This underscores the importance of expanding high-quality experimental datasets to improve generalization, particularly for underrepresented oxidants and VOC classes.

Vreact supports high-throughput screening for emission inventories and atmospheric reactivity assessments. Its applications span VOC prioritization, emission control planning, and kinetic mechanism development, offering actionable insights for environmental policy and modeling. An interactive web interface (http://vreact.envwind.site:8001) (Fig. S3) enhances accessibility for researchers and policymakers. Further improvements in NO<sub>3</sub> and O<sub>3</sub> predictions will expand its utility in nighttime chemistry and secondary aerosol formation scenarios.

# 365 Data and Code Availability

The code and datasets used and/or analysed during the current study are available at https://github.com/Luo-Jiaqi/Vreact and supplemental information.

#### **Supplementary Material**

Detailed information about the learning curve of the Vreact training process (Figure S1); The chemical spatial distribution of VOCs in the OH, O<sub>3</sub>, and NO<sub>3</sub> datasets used in this study and prior literature (Figure S2); User interface of the web platform for predicting VOC reaction rate constants using the Vreact model (Figure S3); Graph representation of molecular structures (Text S1); MPNN message passing and readout phases for molecular graphs (Text S2); Regularization and early stopping techniques in the Vreact model training (Text S3); Model performance evaluation metrics (Text S4); Implementation of the Vreact model (Text S5); Distribution of VOCs reactions with atmospheric oxidants across datasets (Table S1); Atomic features and bond features used in molecular graph representation (Table S2); Hyperparameter search space and optimal settings for

the Vreact model (Table S3); Experimental and predicted  $log_{10}k_i$  values for VOCs on the internal test dataset (Table S4); 447 real-world atmospheric VOCs (Table S5).

# **Author Contributions**

Methodology, Investigation, Formal analysis, Data curation, Visualization, Writing-original draft, X.Z. and J.Q.L; Resources,
Conceptualization, Software, Writing-review & editing, Supervision, Funding acquisition, J.J.F and X.L.; Software,
Validation, Writing-review & editing, W.X.P and Q.X.; Software, Funding acquisition, Writing-review & editing, A.Q.Z;
Resources, Supervision, G.B.J.

# **Competing Interests**

The authors declare no competing interests.

# 385 Financial Support

This research was supported by the Strategic Priority Research Program of the Chinese Academy of Sciences XDB0750100; the project of National Natural Science Foundation of China grant numbers 22193053, 22276197, 22022611 and 92143301; and the Youth Innovation Promotion Association of CAS grant number Y2022020.

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
