# Peer review of "Implications of VOC Oxidation in Atmospheric Chemistry: Development of a Comprehensive AI Model for Predicting Reaction Rate Constants"

_EGUsphere, 2025_

## Author Response (AR1)

**Author's Response**

Dear Editor and Referees,

We greatly appreciate the efforts made by you to improve the quality of our manuscript (MS ID: egusphere-2025-1241). We have carefully reviewed and implemented all the comments provided by you and made significant revisions to the manuscript to address the concerns raised. In this response letter, your comments copied verbatim beneath are in black italic font, the author responses are in normal font, revised text is in blue, and line numbers refer to those in the Track Change manuscript.

**Response to Editor Comments**

Thank you for your submission to Atmospheric Chemistry and Physics. I believe that this manuscript about the estimation of kinetic rate coefficients using a machine learning technique will be very interesting for our readership. Before the manuscript goes into peer-review, I would like the authors to improve on a few aspects as detailed in the following bullet points.

**Response**: Thank you for your thoughtful and constructive comments. We appreciate your recognition of the significance and quality of our work. We have provided detailed responses and made revisions addressing each of the comments.

Background on methodology: The manuscript contains hardly any information about this and similar approaches to estimate reaction rate coefficients or other properties for molecules in the atmospheric sciences. It seems that the methodology of using message-passing neural networks is well established and even part of major packages (e.g. https://keras.io/examples/graph/mpnn-molecular-graphs/), however, this method is not well referenced. You can also refer to similar machine learning studies using graph neural networks (e.g. https://egusphere.copernicus.org/preprints/2025/egusphere-2025-1191/).

**Response:**

We thank the editor for pointing out the need to better contextualize our methodology within the broader landscape of machine learning approaches in atmospheric sciences. In response, we have substantially revised the *Introduction* section to provide a more comprehensive overview of existing methods used for predicting VOC reaction rate constants, including both traditional QSAR models and recent advances in graph-based deep learning.

Specifically, we now describe the limitations of classical experimental and descriptor-based QSAR approaches, and highlight the emergence of graph neural networks (GNNs) as a promising alternative. We cite recent applications of GNNs in atmospheric chemistry and other fields tasks, including GC2NN for vapor pressure prediction (Krüger et al., 2025) and GAT–GIN hybrids for estimating reaction rate constants with •OH (Huang et al., 2024). We further discuss the general MPNN framework (Gilmer et al., 2017), its extensions such as GraphRXN (Li et al., 2023) and Chemprop (Heid et

al., 2024), and clarify that while such models are well established in cheminformatics and synthesis modeling, their use in atmospheric oxidation scenarios remains underexplored.

These additions ensure that our manuscript provides the necessary methodological background and clearly delineates how our contribution advances current approaches.

**The revised content could be found in lines 57-94:**

"Given their importance, accurately predicting the atmospheric oxidation rates of VOCs is critical for understanding their persistence, transformation, and contribution to secondary pollutant formation. Traditionally, such predictions have relied on experimental kinetic modeling methods and quantitative structure-activity relationship (QSAR) approaches (Basant and Gupta, 2018; Liu et al., 2021). Experimental methods involve tracking reactant and product concentrations using techniques like chemical ionization mass spectrometry (CIMS), followed by kinetic fitting to determine Arrhenius parameters (Logan, 1982; Wells et al., 1996). However, these methods are time-consuming and cover only a narrow subset of atmospheric VOCs. QSAR models offer a scalable alternative by leveraging molecular descriptors and statistical learning. Notable examples include AOPWINTM module integrated in US EPI SuiteTM software, which applies Partial Least Squares (PLS) regression to 109 gas-phase reaction with hydroxyl radicals (Atkinson, 1986, 1987; Kwok and Atkinson, 1995), and later expansions using a broader dataset (Öberg, 2005). Some models have also incorporated machine learning algorithms such as multiple linear regression (MLR) (Liu et al., 2020, 2022) for predicting reactions with NO3 and •OH and artificial neural networks for predicting reactions with O3 (Fatemi, 2006). Despite their utility, these models generally rely on predefined descriptors and are typically limited to reactions with a single type of oxidant. Recent advances in deep learning (DL), particularly graph neural networks (GNN), have improved molecular representation by learning features directly from molecular graphs. This enables more flexible and accurate prediction of chemical properties without requiring predefined descriptors. GNNs have been successfully applied in atmospheric chemistry and other fields tasks, such as in predicting vapor pressures with GC2NN (Krüger et al., 2025) and modeling reaction rate constants involving with •OH using GAT-GIN hybrid architectures (Huang et al., 2024). However, like traditional models, these GNN-based frameworks have been developed for single-molecule systems and thus fall short in capturing the complexity of multimolecule reactions in real environments. In contrast, the atmosphere involves competing and sequential reactions between VOCs and multiple oxidants—•OH, NOX, •Cl, and O3—depending on time of day, region, and chemical conditions. This multiplicity underscores the urgent need for models that can simultaneously learn and predict VOC reactivity across multiple oxidants. To meet this need, message passing neural networks (MPNN) offer a powerful framework (Gilmer et al., 2017). MPNNs propagate information across molecular graphs, capturing both atomic-level features and topological context. Extensions of MPNN, such as the communicative GraphRXN (Li et al., 2023) and directed MPNN Chemprop (Heid et al., 2024), have shown promise in learning reactivity across multiple reactants. Yet, their application has largely focused

on synthesis or materials chemistry, not atmospheric multiphase oxidation.

This study addresses this gap by proposing Vreact, a novel Siamese MPNN architecture capable of jointly modelling reactions between VOCs and four major atmospheric oxidants. Unlike previous models that treat each oxidant independently, Vreact processes VOC-oxidant pairs in a unified framework, it learns representations from the molecular graphs of VOCs and oxidants through the MPNN, and encodes their interactions via feature aggregation. This design enables the model to accept arbitrary VOC-oxidant combinations and simultaneously predict reaction rate constants  $k_i$  (where  $i \in \{ \bullet OH, \bullet Cl, NO_3, \text{ or } O_3 \}$ ). Compared to traditional single-oxidant prediction models, Vreact shows significantly improved performance, achieving higher accuracy and and broader generalizability across multiple oxidants. Furthermore, the model's interaction module captures atomic-level interaction patterns, providing mechanistic insights into VOC oxidation process via interpretable interaction weight matrices. Applying Vreact to 447 atmospheric VOCs not included in the training data revealed a wide distribution of oxidation reactivities and confirmed that alkenes and aromatics exhibit higher reactivity, acting as key precursors for ozone and SOA formation."

**The new added references are follows:**

Logan, S. R.: The origin and status of the Arrhenius equation, J. Chem. Educ., 59, 279, https://doi.org/10.1021/ed059p279, 1982.

Wells, R., Baxley, S., and Williams, D.: Rate constants and atmospheric transformations of Air Force VOCs, in: Advanced Technologies for Environmental Monitoring and Remediation, Advanced Technologies for Environmental Monitoring and Remediation, 153–160, https://doi.org/10.1117/12.259768, 1996.

Huang, Z., Yu, J., He, W., Yu, J., Deng, S., Yang, C., Zhu, W., and Shao, X.: AI-enhanced chemical paradigm: From molecular graphs to accurate prediction and mechanism, Journal of Hazardous Materials, 465, 133355, https://doi.org/10.1016/j.jhazmat.2023.133355, 2024.

Krüger, M., Galeazzo, T., Eremets, I., Schmidt, B., Pöschl, U., Shiraiwa, M., and Berkemeier, T.: Improved vapor pressure predictions using group contribution-assisted graph convolutional neural networks (GC2NN), EGUsphere, 1–22, https://doi.org/10.5194/egusphere-2025-1191, 2025.

Li, B., Su, S., Zhu, C., Lin, J., Hu, X., Su, L., Yu, Z., Liao, K., and Chen, H.: A deep learning framework for accurate reaction prediction and its application on high-throughput experimentation data, Journal of Cheminformatics, 15, 72, https://doi.org/10.1186/s13321-023-00732-w, 2023.

Heid, E., Greenman, K. P., Chung, Y., Li, S.-C., Graff, D. E., Vermeire, F. H., Wu, H., Green, W. H., and McGill, C. J.: Chemprop: A Machine Learning Package for Chemical Property Prediction, J. Chem. Inf. Model., 64, 9–17, https://doi.org/10.1021/acs.jcim.3c01250, 2024.

Logarithmic error: From the abstract, it is not clear that the reported MSE relates to the logarithm of the rate coefficient. Please indicate that 0.281 is the error for log(k) or that it is given in log units.

**Response**: We appreciate the reviewer's observation regarding the clarity of the error metrics in the abstract.

In response, we have revised the abstract to explicitly state that the reported MSE of 0.299 and  $R^2$  of 0.941 refer to predictions of  $log_{10}k_i$ , where  $k_i$  denotes the gas-phase

oxidation rate constant of a VOC with oxidants •OH, •Cl, NO3, or O3. The revised sentence now reads:

"The model simultaneously predicts  $\log_{10}k_i$  values and achieves a mean squared error (MSE) of 0.299 and a coefficient of determination (R2) of 0.941 on the internal test set."

Fig. 3: I would have expected that Fig. 3B and 3G show the same data, but this does not seem to be the case as in panel G Vreact scores very well for ozone ( $R^2$  ca. 0.9), but it does not in panel B ( $R^2$  = 0.584). Why is that? In panel C you have to make clear that this is the absolute error of log(k), not the absolute error of k. Overall, panel C is hard to decipher.

**Response**: We thank the reviewer for the careful observation and valuable feedback. The apparent inconsistency between Figure 3B and Figure 3G arises from the fact that they evaluate the model on different datasets and under different setups:

(1) **Figure 3B** shows the performance of the Vreact model on the **internal test set** split from the McGillen et al. dataset, which is a comprehensive compilation of experimentally measured gas-phase reaction rate constants of VOCs with four oxidants (•OH, •Cl, NO3, O3). The R2= 0.584 for O3 corresponds to the model's prediction accuracy on a hold-out set from this unified multi-oxidant dataset.

Figure 3G, on the other hand, compares three different models on literature test datasets for •OH, O3, and •NO3:

- a) The Vreact model (termed *Vreact*) trained on the McGillen dataset and directly applied to external test sets (to assess generalizability),
- b) The Retrained Vreact model (termed *Retrained Vreact*), retrained and evaluated using the same train-test split as in the respective literature (to assess adaptability),
- c) The original published single-oxidant models (Liu et al., Xu et al., etc.). The higher R2 (~0.9) for O3 in Figure 3G reflects the Retrained Vreact model's performance on Xu et al.'s dataset, rather than the original multi-oxidant training setup ( $R^2 = 0.584$ ) shown in 3B. This has now been clarified in both the fig. 3 caption (lines 209-215): "Figure 3. Evaluation and comparison of the predictive performance of the Vreact model. (A) MSE, MAE, R2 of Vreact (trained on the McGillen et al. dataset) on the validation set, internal test set, and external post-2020 test set. (B)  $R^2$  values for  $\log_{10}k_i$  predictions of four oxidants reactions in the internal test set. (C) Distribution of AE between predicted and experimental  $log_{10}k_i$ values for the four oxidants in the internal test set. (D-F) The chemical spatial distribution of VOCs in the •OH, O3, and NO3 datasets used in this study and prior literatures. (G) R2 comparison among previously published single-oxidant models, the original Vreact (evaluated on literature test set), and retrained Vreact (trained and tested using the same splits as the literature models) highlighting adaptability." and the main text (lines 232-244): "Most existing machine learning models for predicting VOC reaction rates constants are tailored for individual oxidants, limiting their applicability to complex atmospheric systems involving multiple oxidants. In contrast, the Siamese MPNN architecture of the Vreact enables simultaneous learning of molecular features and interaction patterns across different

VOC-oxidant pairs within a unified framework. To benchmark Vreact against previously published single-oxidant QSAR/ML models, we selected three top-performing models developed under 298K conditions: Liu et al. (2020) for •OH (180 data points), Xu et al. (2013) for O3 (95 data points), and Liu et al. (2022) for NO3 radicals (189 data points)......Given that our study used different data than those reported in the literature, we employed two strategies for comparison. First, the pre-trained Vreact model (trained on the McGillen dataset) was directly applied to the test sets from the literature to evaluate extrapolation performance. Second, Vreact was retrained on each literature dataset using their original train/test splits, allowing a direct comparison with published models on literature test sets (Retrained Vreact)."

(2) We also thank the reviewer for pointing out that the clearly labeled shown in Fig. 3C. We have now revised the figure axis label to state clearly that it shows the absolute error of  $\log_{10}k_i$ , not  $k_i$ . The revised figure is as follows:

Online tool: I commend the authors for putting their code and tool online, this will be very much valued by the community. However, in my testing, I could not get a value returned by the web tool. Even after minutes of waiting, it still says "running". Can this be fixed or explained?

**Response**: Thank you for your comment. We apologize that the web tool was not working properly due to a port malfunction, we fixed the web tool and replaced the port with a new one, the web tool is now at the following: <a href="http://vreact.envwind.site:8001/">http://vreact.envwind.site:8001/</a>. The web tool is already working.

Formatting of references: Please insert a space before each reference, e.g. l. 26 "prediction (Abramson et al., 2024)". This must be a relic from the conversion from a previously submitted version.

**Response**: We appreciate the reviewer bringing up this important point. We have inserted a space before each reference, *e.g.* line 27 of the revised manuscript "molecular generation (Zhang et al., 2023)".

Formatting: Consider using a lower-case k for reaction rates as upper case K could be mistaken for equilibrium constants - what does "i" stand for in K i?

**Response**: Thank you for the reviewer's insightful comments. We have changed the upper case "K" to a lower case "k" to indicate the reaction rates. The "i" in  $k_i$  represents the four oxidants ( $\bullet$ OH,  $\bullet$ Cl, NO3 or O3), e.g.,  $k_{\bullet}$ OH represents the reaction rate for the reaction of VOCs with  $\bullet$ OH. We have added an explanation of "i" in line 88 of the revised manuscript: "reaction rate constants  $k_i$  (where  $i \in \{\bullet$ OH,  $\bullet$ Cl, NO3, or O3 $\}$ )".

**Response to Referee #1**

**General comments**

The study in question presents a new model for the prediction of reaction rate constants of volatile organic compounds (VOCs). The authors used the reaction rate constant dataset by McGillen et al. to train a Siamese message passing neural network (MPNN) to predict these rate constants. The outcoming model was given the name "Vreact" and it was shown to outperform existing models for reaction rate constant prediction. The dataset used in this study comprises 2802 gas-phase reaction rate constants for 1586 VOCs and 4 oxidants (•OH, •Cl, •NO3 and O3). The authors underline this diversity of oxidants as one of their advantages compared to previous models which only use a single oxidant per model. Because of the wide value range of reaction constants, the values were log-transformed. Vreact takes the SMILES string of the VOC and the oxidant as inputs, which is an established and modern approach in cheminformatics. Graph representations are generated from these inputs and fed to the neural network that creates the molecular feature tensors A and B. Further mathematical operations are executed to account for the effects of molecular interactions. Finally, the prediction value for the reaction rate constant is made.

Moreover, the authors evaluate how Vreact can contribute to the understanding of aerosol formation mechanisms. They showcase the oxidation of 2-methyl-4-penten-2-ol, discussing different reaction pathways and how the interaction layer of Vreact can be used for comprehension. Furthermore, the authors gathered more data from 2020 and onwards, which they called the 'post-2020 test set' to analyze the extrapolation ability of Vreact, leading to satisfactory results. Besides, more insights on the reaction rates of specific chemical classes are provided.

All in all, the article presents a modern and sustainable study. The Vreact model that is the key component of this work was built on well-established methods and principles and could overall convince with its performance. Vreact's advantages and improvements towards other models were clearly outlined in a comprehensible way. The study was conducted scientifically correct with no obvious shortcomings. Despite it being a rather data scientific topic, its atmospheric relevance became evident. The illustrations used are helpful and supporting. The supplementary material contains further details on the model architecture and is useful for a deeper understanding. Another valuable resource is the web tool version of Vreact, reinforcing reproducibility and open data.

**Response**: Thank you for your thoughtful and constructive comments. We appreciate your recognition of the significance and quality of our work. We have provided detailed responses and made revisions addressing your comments.

**Specific comments**

After the results of the test set were presented, the authors provided more extensive evaluations and showcases of the model's abilities. First, they draw a more detailed comparison between Vreact and the existing single-oxidant models. Therefore, they use two independent approaches: 1) using the pre-trained Vreact to predict the test sets

from the literature and 2) retraining Vreact on the original train/test splits of the literature. Approach 2) is a bullet-proof method that really isolates the model's predictive capability and delivers a nice comparison. Approach 1) has the potential problem, that the literature test sets contain data points that are part of Vreact's training set. This would be problematic, because generally, machine learning models perform significantly better on seen data, resulting in an unfair comparison. It would be appreciated, if the authors could address this issue briefly, since it was unmentioned in the text so far.

**Response:**

We thank the reviewer for highlighting the issue in Approach 1 regarding the partial overlap between the literature test sets and the Vreact training set, which could lead to an unfair performance comparison. We identified and removed the duplicate molecules (2 of 38 for NO3, 13 of 35 for O3, and 6 of 36 for OH) from the literature test sets. The revised R2 values were recalculated and are now presented in the updated Figure 3G. While the R2 values have decreased slightly (OH:0.024/O3:0.016/NO3:0.027), the overall comparative trends remain unchanged. To enhance clarity, we now refer to the modified test sets as "cleaned literature test sets" and the original ones as "original literature test sets" throughout the revised manuscript.

(G) R2 comparison among previously published single-oxidant models, the original Vreact (evaluated on cleaned literature test sets), and Retrained Vreact (trained and tested using the same original splits as the literature models) highlighting adaptability.

**The revised main text lines 232-241:**

The original text "Liu et al. (2020) for •OH (180 data points), Xu et al. (2013) for O3 (95 data points), and Liu et al. (2022) for NO3 radicals (189 data points). " has been revised to "Liu et al. (2020) for OH (training/test = 144/36), Xu et al. (2013) for O3 (60/35), and Liu et al. (2022) for NO3 radicals (151/38)."

"To ensure a fair comparison, overlapping data points between the literature test sets and the McGillen training set were removed (2 of 38 for NO3, 13 of 35 for O3, and 6 of 36 for OH). Second, Vreact was retrained on each literature dataset using their original train/test splits (Retrained Vreact), allowing a direct comparison with published models

on original literature test sets." has been added.

**Technical corrections**

No typing errors or other technical problems were found.

Response: Thank you for checking for typing errors or other technical problems.

**Response to Referee #2**

Zhang and co-authors present a machine learning model for predicting reaction rate constants of VOC-oxidant pairs using a Siamese neural network. The model is novel in its design and application combination, especially in handling multiple atmospheric oxidants. The results demonstrate good predictive performance alongside chemical insight. The results also demonstrate varying performance on the test set depending on which oxidant is considered. The model is tested on an additional external dataset, and is used to make predictions of rate constants for compounds lacking measurements.

From my point of view, the manuscript is generally well-written and clearly structured. However, methodological and interpretative aspects would benefit from clarification to ensure reproducibility and help contextualize the findings. However, I happily recommend it for publication subject to minor revision.

**Response**: Thank you for your thoughtful and constructive comments. We appreciate your recognition of the significance and quality of our work. We have provided detailed responses and made revisions addressing your comments.

**General comments**

1.I understand that the major benefit of Vreact is the ability to predict reactivities for multiple oxidants. Could the authors further clarify the motivation for using a Siamese neural network over simpler alternative architechtures which also could provide prediction for multiple oxidants (such as a one-hot encoding of oxidant identity). Given that only four oxidants are included, it would be helpful to understand whether the architecture was chosen for scalability, improved interpretability, or flexibility. Will more oxidants be considered in the future?

**Response**: Thank you for the insightful comments. As you mentioned, simpler architectures can indeed provide predictions for multiple oxidants simultaneously. We chose the Siamese architecture for the following reasons:

- 1. Flexibility: The Siamese GNN architecture used in Vreact possesses the flexibility inherent to deep learning, which simple one-hot encoding/machine learning models lack. Because oxidants are treated as molecules rather than abstract categories, the model can leverage structural similarities between known and novel oxidants to transfer learned interaction patterns. This is particularly important in atmospheric chemistry, where newly identified or understudied oxidants may be structurally or electronically related to those in the training set.
- **2. Interpretability:** The pairwise design enables the extraction of interaction matrices between atoms of VOCs and oxidants, which can be visualized and interpreted (Figure 4). This level of interpretability would not be possible in architectures where the oxidant is reduced to a categorical token, and it provides mechanistic insights into reactive sites and molecular interactions.
- **3. Scalability:** The Siamese GNN architecture of Vreact enhances its scalability. For simple one-hot encoding/machine learning models, designed descriptors/molecular fingerprints are required for the research objects. However, there are numerous reactions in the atmosphere with diverse mechanisms. Requiring a simple architecture

to be applicable to non-research objects will affect its interpretability and predictive performance. Therefore, the scalability of a simple architecture is very limited. The Siamese GNN used in Vreact does not rely on predefined descriptors/molecular fingerprints but performs end-to-end modeling. This architecture grants Vreact the ability to expand to other oxidants. Currently, we have only considered four oxidants because these four are widely studied and have sufficient data, which facilitates modeling. In the future, if higher-quality and more extensive datasets become available, we will incorporate more oxidants and update the website in a timely manner.

In the main text, we further elaborate on the limitations of other methods and the advantages of Vreact:

"Despite their utility, these models generally rely on predefined descriptors and are typically limited to reactions with a single type of oxidant, which constrains the scalability of the model." (Line 76 in the revised manuscript)

"Extensions of MPNN, such as the communicative GraphRXN (Li et al., 2023) and directed MPNN Chemprop (Heid et al., 2024), have shown promise in learning reactivity across multiple reactants. They extract the interaction features of chemical reactions in depth, rather than performing simple reactant concatenating. Yet, their application has largely focused on synthesis or materials chemistry, not atmospheric multiphase oxidation." (Lines 88-89 in the revised manuscript)

"Compared to traditional and simple single-oxidant prediction models, Vreact shows significantly improved performance, achieving higher accuracy, stronger interpretability and wider scalability across multiple oxidants. Furthermore, based on the flexibility of the DL architecture, the designed interaction module captures atomic-level interaction patterns, providing mechanistic insights into VOC oxidation process *via* interpretable interaction weight matrices." (Lines 96-98 in the revised manuscript)

2.A brief discussion of quantum chemistry methods to compute these types of rate constants is not mentioned in the background, but could help position this new method in the broader context of rate constant prediction for atmospheric reactions.

**Response**: Thank you for the comment. We appreciate this suggestion and have now added a concise overview of quantum-chemical (QC) approaches that are widely used to estimate gas-phase rate constants for atmospheric reactions.

The new paragraph (see additions below) highlights both the strengths and limitations of quantum chemistry (QC) methods. This context helps position Vreact as a complementary, data-driven alternative that can deliver near-instant predictions for thousands of VOC–oxidant pairs while retaining mechanistic interpretability. "Traditionally, such predictions have determined either through experimental kinetic modeling methods, (Basant and Gupta, 2018; Liu et al., 2021). which track reactant and product concentrations using techniques such as chemical ionization mass spectrometry (CIMS) and apply kinetic fitting to derive Arrhenius parameters (Logan, 1982; Wells et al., 1996), or through computational methods based on high-level quantum chemical calculations that simulate reaction pathways and energy barriers. However, these methods are time-consuming and cover only a narrow subset of atmospheric VOCs.

"...a narrow subset of atmospheric VOCs. QC approaches combine ab initio or density-functional theory calculations with transition-state theory (TST), canonical or variational TST to obtain temperature-dependent rate constants (Canneaux et al., 2014; Liu et al., 2021; Meana-Pañeda et al., 2024). While QC methods offer detailed mechanistic insight, their computational cost scales steeply with molecular size and conformational complexity, limiting routine application to large numbers of VOCs. However, traditional computational methods have shortcomings such as high computational complexity and low efficiency. As a more scalable alternative, QSAR model leverage molecular descriptors and statistical learning, and it has become one of the important methods for evaluating reaction rate constants." (Lines 64-70 in the revised manuscript)

3.Methods and Table S1 suggest that stratified sampling was used to balance oxidant classes across train/validation/test splits. Since the model operates on VOC–oxidant pairs, it is now unclear whether the same VOC can appear in different splits with different oxidants. If so, this could introduce information leakage. Please clarify whether VOCs were kept disjoint across splits.

**Response**: We thank the reviewer for raising this important concern. In our study, stratified sampling was performed on VOC–oxidant pairs, which means that the same VOC may appear in different data splits when paired with different oxidants. In other words, VOCs were not kept disjoint across splits.

We acknowledge the potential concern regarding information leakage. However, we believe that the current design is appropriate for the following reasons:

- 1. **Model design focus**: Vreact is trained to model interactions between VOCs and oxidants, not the VOCs alone. Each VOC-oxidant pair represents a distinct chemical reaction, and the underlying mechanisms often vary substantially across oxidants. Thus, each pair can be considered a unique input, and allowing the same VOC to appear with different oxidants across splits does not constitute classical data leakage.
- 2. **Chemical diversity**: Forcing the same VOC to appear only in one split (*e.g.*, training only) would eliminate the number of VOC-oxidant combinations, reducing the diversity and coverage of the training set for model learning.
- 3. **Empirical performance**: If leakage were present, it would likely result in inflated test performance. However, as shown in our results, the model maintains strong generalization and extrapolation performance, including on unseen VOCs and external test sets, suggesting that overfitting due to repeated VOCs is not a concern.

To clarify this in the manuscript, we have added the following statement:

"Combinations of the same VOC with different oxidants may appear across the training, validation, and internal test sets." has been added to the main text. (Lines 112-113 in the revised manuscript)

4.In Figure 3G, model performance on the external OH dataset is lower than for  $O_3$ , which is the reverse of the trend observed in the internal test set. Could this difference be a result of data quality, compound overlap, or target range?

**Response**: Thank you for this thoughtful comment. We agree that the inverse performance trend observed between OH and O3 in the internal versus external (literature) test sets warrants further clarification.

We have carefully examined this discrepancy and found that it is primarily attributable to differences in **chemical space coverage**, rather than data quality or compound overlap alone. Specifically:

All external test sets are sourced from literatures and the data quality is reliable. Any duplicate pairs between the training data and the literature test sets were removed prior to evaluation.

The internal OH test set includes many VOCs with broad representation and strong overlap with the training set, resulting in high performance. In contrast, the external OH test set contains only 36 VOCs, many of which are sparsely distributed near the periphery of the training data distribution (Fig. 3E). This leads to a moderate drop in R2 despite the model's generally strong performance for OH.

For O3, however, the internal test set includes structurally atypical compounds that are distant from dense the training data in latent space (Fig. 3F, right region), resulting in a lower R2. The literature O3 test set, by contrast, is more clustered and lies closer to the training set in chemical space, allowing the model to achieve higher R2 despite the small sample size.

This difference is analyzed in the manuscript:

"Notably, Vreact shows opposite performance trends for OH and O3 between the internal and literature test set. To understand this, UMAP was applied to project compounds from the training, internal, and literature test sets into a shared chemical space. As shown in Fig. 3E, the internal OH test set overlaps well with the training data, leading to consistently strong performance. In contrast, the literature OH set is sparse and scattered near the dataset boundaries. Despite this, Vreact still achieves a high R², demonstrating good generalization. For O3 (Fig. 3F), the internal test set lies farther from the dense training distribution, contributing to lower R². Meanwhile, the literature O3 set is better aligned with the training data, resulting in higher prediction accuracy. For NO3 (Fig. 3G), both internal and literature sets show similar distributions, and the model achieves comparable R² values (~0.815). Although Vreact underperforms slightly compared to the original single-oxidant model, retraining on the literature data improves performance. This suggests that multi-oxidant training may introduce some noise but does not significantly compromise prediction accuracy." (Lines 275-291 in the revised manuscript)

The previous Figure 3D-3F has been modified to Figure S2:

Figure S2. The chemical spatial distribution of VOCs in the OH, O3, and NO3 datasets used in this study and prior literatures.

Figure 3D-3F has been modified to Figure 3E-3G:

(E-G) The chemical spatial distribution of VOCs in the OH, O3, and NO3 datasets used in this study and prior literatures.

5.Clustering was used to analyze molecular groups and their reactivity (Figures 2E, 3D–F), but details on how these embeddings and clusters were generated are not provided in methods. It would be good with a brief description of how the morgan fingerprint was constructed (which parameters) in the methods. Similarly, UMAP and the SOM methods could be briefly described, along with any hyperparameters, in the methods.

**Response**: Thank you for your rigorous suggestion. We agree that further clarification of the clustering methodology is necessary and have added relevant descriptions to the Methods section.

The Methods section now includes this information as follows::

**"2.3 Clustering analysis**

Morgan fingerprints (radius 2, 1024 bits, generated using RDKit) was used as the molecular embeddings before clustering and visualization. To investigate VOC structural diversity and reactivity trends, two methods were applied: the Self-Organizing Map (SOM) (Kohonen, 2006) and the Uniform Manifold Approximation and Projection (UMAP). The SOM algorithm clustered VOCs into 100 structural groups (10×10 grid), using a sigma of 0.3 and learning rate of 0.5. The UMAP algorithm projected the high-dimensional fingerprint space into 2D for visualization, with the number of neighbors set to 50, minimum distance to 0.6, and metric as correlation." (Lines 150-155 in the revised manuscript)

Additionally, the original text describing SOM clustering in Fig. 2E was revised for clarity:

"The SOM algorithm was used to explore the relationship between VOC structural

variation and  $\log_{10}k_i$ . Each grid in Fig. 2E represents a VOC cluster, and the color gradient indicates reactivity (average  $\log_{10}k_i$  values) for the corresponding oxidants. By comparing  $\log_{10}k_i$  values across clusters, oxidant-specific reactivity patterns can be assessed." (Lines 190-193 in the revised manuscript)

6. Finally, the manuscripts would benefit from an outlook contextualizing the model's performance by identifying which applications the current accuracy supports and which may require improvement. Relating how performance varies across different oxidants and how this relates to the amount of available data could further emphasize the paper's contribution to understanding data requirements for reliable model accuracy for atmospheric applications.

**Response**: Thank you for the valuable suggestion. We have added a dedicated outlook section in the revised manuscript to contextualize the model's performance, discuss its applicability across atmospheric chemistry tasks, and highlight the relationship between accuracy and data availability. These revisions are incorporated into the *Concluding* section (Lines 338-351) to clarify the model's application scope and remaining challenges.

"In response to growing concerns about atmospheric pollution and its impact on human health and climate, this study introduces Vreact, a deep learning model designed to predict oxidation rate constants for VOCs with multiple oxidants (OH, Cl, NO3, O3). Vreact demonstrates strong overall performance (MSE=0.299, R2=0.941 on internal test data) and provides mechanistic insights by capturing atomic-level interaction patterns through a Siamese MPNN framework. Its predictive accuracy varies by oxidant, reflecting the availability and diversity of training data. The model achieves high accuracy for OH (R2=0.929, n=1363) and Cl (R2=0.913, n=735), supporting robust application in daytime oxidation modeling. In contrast, lower performance is observed for NO3 (R2=0.721, n=393) and O3 (R2=0.584, n=311), pointing to challenges in modeling oxidants with fewer data and more complex mechanisms. This underscores the importance of expanding high-quality experimental datasets to improve generalization, particularly for underrepresented oxidants and VOC classes.

Vreact supports high-throughput screening for emission inventories and atmospheric reactivity assessments. Its applications span VOC prioritization, emission control planning, and kinetic mechanism development, offering actionable insights for environmental policy and modeling. An interactive web interface (<a href="http://vreact.envwind.site:8001">http://vreact.envwind.site:8001</a>) (Fig. S3) enhances accessibility for researchers and policymakers. Further improvements in NO3 and O3 predictions will expand its utility in nighttime chemistry and secondary aerosol formation scenarios."

**Specific comments**

1.Line 29: Add citations on data-driven methods applied to atmospheric chemistry.

**Response**: Thank you for your valuable feedback. We have added 4 citations on data-driven methods applied to atmospheric chemistry in line 29.

"...Environmental challenges, particularly those associated with atmospheric chemistry and climate change (Chen et al., 2024; Kubečka et al., 2023; Qiu et al., 2023; Zhao et

al., 2025),..."

Chen, X., Ma, W., Zheng, F., Wang, Z., Hua, C., Li, Y., Wu, J., Li, B., Jiang, J., Yan, C., Petäjä, T., Bianchi, F., Kerminen, V.-M., Worsnop, D. R., Liu, Y., Xia, M., and Kulmala, M.: Identifying Driving Factors of Atmospheric N2O5 with Machine Learning, Environ. Sci. Technol., 58, 11568–11577, https://doi.org/10.1021/acs.est.4c00651, 2024.

Kubečka, J., Knattrup, Y., Engsvang, M., Jensen, A. B., Ayoubi, D., Wu, H., Christiansen, O., and Elm, J.: Current and future machine learning approaches for modeling atmospheric cluster formation, Nat. Comput. Sci., 3, 495–503, https://doi.org/10.1038/s43588-023-00435-0, 2023.

Qiu, Y., Feng, J., Zhang, Z., Zhao, X., Li, Z., Ma, Z., Liu, R., and Zhu, J.: Regional aerosol forecasts based on deep learning and numerical weather prediction, npj Clim. Atmos. Sci., 6, 71, https://doi.org/10.1038/s41612-023-00397-0, 2023.

Zhao, Y., Zheng, B., Saunois, M., Ciais, P., Hegglin, M. I., Lu, S., Li, Y., and Bousquet, P.: Air pollution modulates trends and variability of the global methane budget, Nature, 642, 369–375, https://doi.org/10.1038/s41586-025-09004-z, 2025.

2.Line 45: "primarily"  $\rightarrow$  "primary."

Response: Thanks for your comment. The "primarily" has been replaced with "primary".

3.Line 48: The phrase "with NO3 radicals" is repeated—I suggest to remove one instance.

**Response**: Thanks for your valuable comment. The repeated phrase "with NO3 radicals" has been replaced.

4.Line 48: "the atmosphere's self-cleaning capacity" is ambiguous; consider clarifying. rephrasing or removing.

**Response**: Thank you for pointing out this problem in manuscript. The "...significantly influencing the spatial and temporal variation of the atmosphere's self-cleaning capacity and the formation of organic aerosols." has been modified to "...significantly influencing the spatial and temporal variation of the formation of organic aerosols."

5.Line 90: Typo— "and and"

**Response**: Thanks for your constructive suggestion. The repeated "and" has been removed.

6.Line 149: "functional group" could be replaced with "molecular motif" when referring to double bonds.

**Response**: Thank you for the suggestion. The "including 22 functional groups" has been modified to "including 22 molecular motifs". The "(B) Number of VOCs containing each functional group" has been modified to "(B) Number of VOCs containing each molecular motif".

7.Line 191– It is mentioned in results that MSE is the metric that was used for hyperparameter optimization. This information should also be included in the Methods

section for clarity.

**Response**: We are very sorry for our negligence. We have included the information you mentioned in the Methods section for clarity. The "After identifying the optimal hyperparameter combination (Table S3) on the validation set, and the best model was saved" has been modified to "During hyperparameter optimization, the hyperparameter combination that minimizes the Mean Squared Error (MSE) of the validation set was selected as the optimal hyperparameter combination, and the best model was saved (Table S3)".

8.Improve resolution of Figures 1–5. Figure 5A would be clearer as a conventional bar chart rather than a circular one for better being able to match bar height with y value. **Response**: Thanks for you kindly comment. We have improved the resolution of Figures 1-5 and modified Figure 5A into a conventional bar chart.

Finally, we would like to thank you again for your great efforts on improving the quality of this manuscript.

Thank you all, Yours sincerely, Xian Liu

---

## Referee Report (RR1)

Review for "Implications of VOC Oxidation in Atmospheric Chemistry: Development of a comprehensive AI Model for Predicting Reaction Rate Constants" by Xin Zhang, Jiaqi Luo, Wenxiao Pan, Qiao Xue, Xian Liu, Jianjie Fu, Aiqian Zhang, Guibin Jiang

**General comments**

The study in question presents a new model for the prediction of reaction rate constants of volatile organic compounds (VOCs). The authors used the reaction rate constant dataset by McGillen et al. to train a Siamese message passing neural network (MPNN) to predict these rate constants. The outcoming model was given the name "Vreact" and it was shown to outperform existing models for reaction rate constant prediction.

The dataset used in this study comprises 2802 gas-phase reaction rate constants for 1586 VOCs and 4 oxidants ( $\cdot$ OH,  $\cdot$ Cl,  $\cdot$ NO3 and O3). The authors underline this diversity of oxidants as one of their advantages compared to previous models which only use a single oxidant per model. Because of the wide value range of reaction constants, the values were log-transformed. Vreact takes the SMILES string of the VOC and the oxidant as inputs, which is an established and modern approach in cheminformatics. Graph representations are generated from these inputs and fed to the neural network that creates the molecular feature tensors A and B. Further mathematical operations are executed to account for the effects of molecular interactions. Finally, the prediction value for the reaction rate constant is made.

Moreover, the authors evaluate how Vreact can contribute to the understanding of aerosol formation mechanisms. They showcase the oxidation of 2-methyl-4-penten-2-ol, discussing different reaction pathways and how the interaction layer of Vreact can be used for comprehension. Furthermore, the authors gathered more data from 2020 and onwards, which they called the 'post-2020 test set' to analyze the extrapolation ability of Vreact, leading to satisfactory results. Besides, more insights on the reaction rates of specific chemical classes are provided.

All in all, the article presents a modern and sustainable study. The Vreact model that is the key component of this work was built on well-established methods and principles and could overall convince with its performance. Vreact's advantages and improvements towards other models were clearly outlined in a comprehensible way. The study was conducted scientifically correct with no obvious shortcomings. Despite it being a rather data scientific topic, its atmospheric relevance became evident. The illustrations used are helpful and supporting. The supplementary material contains further details on the model architecture and is useful for a deeper understanding. Another valuable resource is the web tool version of Vreact, reinforcing reproducibility and open data.

**Specific comments**

The revised version does not contain any problematic issues to be discussed here.

**Technical corrections**

No typing errors or other technical problems were found.

---

## Author Response (AR2)

**Response to Editor Comments**

Dear Editor,

We greatly appreciate the efforts made by you to improve the quality of our manuscript (MS ID: egusphere-2025-1241). We have carefully reviewed and implemented all the comments provided by you and made significant revisions to the manuscript to address the concerns raised. In this response letter, your comments copied verbatim beneath are in black italic font, the author responses are in normal font, revised text is in blue, and line numbers refer to those in the Track Change manuscript.

Thank you very much for your revision. Before the paper can be accepted in ACP, I would like to reiterate on an important point made by reviewer #2, which I believe was not sufficiently addressed and is crucial to show the significance of this work. The original comment read:

"I understand that the major benefit of Vreact is the ability to predict reactivities for multiple oxidants. Could the authors further clarify the motivation for using a Siamese neural network over simpler alternative architechtures which also could provide prediction for multiple oxidants (such as a one-hot encoding of oxidant identity). Given that only four oxidants are included, it would be helpful to understand whether the architecture was chosen for scalability, improved interpretability, or flexibility."

I would prefer if the authors give quantitative results for their newly added statement of "achieving higher accuracy, stronger interpretability and wider scalability".

- 1. Accuracy: It would really strengthen the paper if the authors could show and quantify, using unbiased calculations (e.g. with same training time), that the Siamese neural network architecture is superior to simpler alternative model architectures. Can the authors show what happens if you train the same type of neural network (just without the Siamese architecture), for each oxidant? Is the result worse? I think it's not enough here to compare with other published models that may have different numbers of neural network hyperparameters, training time, etc.
- 2. Interpretability: Can the authors show what precisely can be learned about the detailed chemical interactions of two molecules with the Siamese neural network architecture? If not, it should be indicated that this is only a hypothetical feature that was not yet explored.

**Response**: We thank the reviewer for the careful observation and valuable feedback.

**1. Accuracy:**

To quantitatively demonstrate the benefits of the Siamese neural network architecture under identical training conditions, we conducted an ablation study in the revised manuscript. In the **Vreact-Ablation** model, we removed the oxidant input and interaction module from the original Vreact architecture, retaining only the VOCs input. This results in a simpler single-input MPNN architecture, while keeping all other settings—including training/validation/test splits, data source (McGillen et al., 2020), hyperparameters, training time, and evaluation metrics—

identical to those of Vreact. We trained both models on individual oxidant datasets (OH, Cl, NO3, O3) as well as on the combined dataset. As shown in **Fig. 3D**, Vreact consistently outperformed Vreact-Ablation across all oxidants.

- The improvements of R2 for OH, Cl, NO3, and O3 are 0.049, 0.113, 0.184, and 0.021 respectively.
- When evaluated on the full dataset (all oxidants), Vreact-Ablation achieved an R2 of only 0.035, indicating that the simple MPNN without oxidant information fails to generalize beyond single-oxidant learning.
- Moreover, their runtime per iteration is essentially consistent.

**We have added the following content:**

Figure 3. Evaluation and comparison of the predictive performance of the Vreact model. (A) MSE, MAE,  $R^2$  of Vreact (trained on the McGillen et al. dataset) on the validation set, internal test set, and external post-2020 test set. (B)  $R^2$  values for  $\log_{10}k_i$  predictions of four oxidants' reactions in the internal test set. (C) Distribution of AE between predicted and experimental  $\log_{10}k_i$  values for the four oxidants in the internal test set. (D)  $R^2$  of the Vreact and Vreact-Ablation on the OH, Cl, NO3, O3, and combined test sets. (E)  $R^2$  comparison among previously published single-oxidant models, the original Vreact (evaluated on cleaned

literature test sets), and Retrained Vreact (trained and tested using the same original splits as the literature) highlighting adaptability. (F-H) The chemical spatial distribution of VOCs in the OH, O3, and NO3 datasets used in this study and prior literature.

**"3.3 Model Ablation Study**

To evaluate the contribution of the Siamese neural network architecture in Vreact, we performed an ablation study. In the ablation model (Vreact-Ablation), the oxidant input and interaction module were removed, leaving only the VOC input. Both Vreact and Vreact-Ablation were trained, validated, and tested on the OH, Cl, NO3, O3, and combined datasets. All experimental settings were kept consistent, including data sources (McGillen et al., 2020), hyperparameters and evaluation metrics. As shown in Fig. 3D, Vreact consistently outperformed Vreact-Ablation across all four oxidants, with R2 improvements of 0.049 (OH), 0.113 (Cl), 0.184 (NO3), and 0.021 (O3). When evaluated on the combined dataset, Vreact-Ablation achieved an R2 of only 0.035, indicating that it fails to generalize across multiple oxidants. Additionally, both models showed comparable runtime per iteration. These results demonstrate that, under the same training conditions, the Siamese MPNN architecture significantly enhances predictive performance and generalization. By explicitly modeling VOC-oxidant interactions, the architecture enables the network to capture shared patterns across reaction types, thereby improving its practical applicability in multi-reactivity prediction."

**2. Interpretability:**

Regarding interpretability, Vreact's architecture was specifically designed to capture atomic-level interactions between VOCs and oxidants. The model incorporates an interaction module that learns cross-molecular attention weights, enabling insights into which atoms in each molecule contribute most to reactivity.

In Section 3.4, we present visualization and analysis of these interaction weights, illustrating how the model differentiates chemical behaviors of VOCs toward different oxidants. While still preliminary, this direction demonstrates the architectural potential for interpretable chemical learning. We have clarified that this aspect is explicitly explored, not merely hypothetical.

l.64: "QC approaches combine ab initio or density-functional theory calculations ..."DFT is generally considered an ab initio method (though not a first-principlesmethod).

**Response:** Thank you for your comment. "QC approaches combine ab initio or density-functional theory calculations with transition-state theory (TST), canonical or variational TST to obtain temperature-dependent rate constants" has been modified to "QC approaches use density-functional theory calculations such as transition-state theory (TST) or variational TST to obtain temperature-dependent rate constants".

1.90: "They extract the interaction features of chemical reactions in depth, rather than performing simple reactant concatenating. Yet, their application has largely focused on synthesis or materials chemistry, not atmospheric multiphase oxidation."

- Can the authors explain what they mean with the first sentence?
- Second sentence: this study does not look at multiphase chemistry, either.

**Response:** Thank you for the comments. The following is an explanation of your comments one by one.

- First sentence: We are very sorry that this sentence has caused you confusion. We would like to show through two examples that GNN and especially MPNN have been deeply applied when modeling chemical reactions. These two methods can transform molecules into molecular graphs, then input them into MPNN for feature depth extraction, and finally splice the features for reaction prediction. They do not perform simple concatenation for prediction by converting reactants into molecular fingerprints/descriptors. The MPNN provides more chemical information for chemical reaction modeling. So, the "They extract the interaction features of chemical reactions in depth, rather than performing simple reactant concatenating" has been modified to "Compared with the simple concatenation using molecular fingerprints/descriptors, they all use MPNN to deeply extract task-relevant representations of chemical reactions, provide abundant chemical information for subsequent reaction modeling, and achieve good prediction results".
- Second sentence: We apologize for the inappropriate expression of the second sentence. Our study indeed does not look at multiphase chemistry. The "atmospheric multiphase oxidation" has been modified to "atmospheric oxidation reaction".

1.98: "Vreact shows significantly improved performance"

- Please provide the level of significance for this statement

**Response:** Thank you for the valuable suggestion. In the ablation experiment, Vreact outperforms Vreact-Ablation in predictive performance across all four oxidants, with R2 differences between the two models on OH, Cl, NO3, and O3 being 0.049, 0.113, 0.184, and 0.021, respectively. Additionally, when both models are tested using the combined dataset, Vreact-Ablation achieves an R2 of only 0.035, indicating that it lacks predictive capability. Vreact shows significantly improved performance compared to simpler alternative architectures. Then, compared with the simple single-oxidant prediction models reported in three literature, Vreact exhibits predictive performance that is either superior to or comparable with those models. Vreact did not demonstrate significantly improved prediction performance on the NO3 literature test set.

We have removed the imprecise statement: "Compared to traditional and simple single-oxidant prediction models, Vreact shows significantly improved performance, achieving higher accuracy, stronger interpretability and wider scalability across multiple oxidants".

**Add new statements:**

"The dual-input design of Vreact enhances scalability and generalization across multiple oxidants. Ablation experiments show that Vreact significantly outperforms a structurally simpler single-input MPNN trained under identical conditions. The interaction module within Vreact provides atomic-level attention maps that offer

mechanistic insights into VOC-oxidant reactivity patterns, improving interpretability." (Lines 97-100)

Finally, we would like to thank you again for your great efforts on improving the quality of this manuscript.

Thank you all, Yours sincerely, Xian Liu

---

## Author Response (AR3)

Dear Editor,

Thank you so much for your kind email and the wonderful news of accepting our paper for publication. We are truly grateful for the opportunity to contribute to the journal, and we greatly appreciate the thorough and constructive feedback provided during the peer review process-it has significantly helped improve the quality of our work.

Once again, thank you for your support and recognition.

Thank you all, Yours sincerely, Xian Liu